# Reproducibility Study of "FairCLIP: Harnessing Fairness in Vision-Language Learning"

## Abstract

Fairness is a crucial consideration in medical deep learning, as model bias can lead to disparities in diagnoses and treatment decisions. Luo et al. (2024a) conducted a comprehensive fairness analysis of two vision-language models, CLIP and BLIP2, revealing significant bias in their predictions. The authors introduced FairCLIP, a model that mitigates bias and achieves a better performance-fairness trade-off. In this work, we aim to (1) reproduce the key findings of Luo et al. (2024a) and (2) extend their analysis with additional evaluations. Our results confirm that most of the reported findings are reproducible, although we identify discrepancies in specific cases. Furthermore, we conduct a more extensive fairness analysis by incorporating two additional metrics: Precision Disparity and Mean Absolute Deviation. Following this analysis, we confirm the presence of bias in CLIP. However, despite being able to reproduce most of the results, we challenge the claim that FairCLIP improves fairness. Our results suggest that improvements of FairCLIP over CLIP are inconsistent and architecture- or attribute-dependent, rather than a generalizable improvement in fairness. Finally, we conduct a study to identify the source of bias. Our results indicate that the bias does not originate from the summarized clinical notes, medical pre-training or group imbalance.

## 1 Introduction

Artificial intelligence (AI) is expected to broadly transform medicine, improving the experiences of both clinicians and patients (Rajpurkar et al., 2022). Although relatively few AI tools have been translated into medical practice (Wiens et al., 2019), AI has been shown to be successful in a large variety of retrospective studies. This success has been particularly evident in medical domains that rely on the interpretation of images, driven by deep learning advances in image classification. AI has made significant contributions to, for instance, ophthalmology (Liu et al., 2019; Milea et al., 2020) and radiology (McKinney et al., 2020; Ardila et al., 2019).

Still, challenges to medical AI research remain, like the cost of obtaining large, labeled datasets in healthcare (Ghesu et al., 2022). Foundation models trained with self-supervised learning (SSL) methods address this issue. By enabling models to learn meaningful representations directly from unlabeled data (Khan et al., 2023), SSL reduces the reliance on extensive manual annotation, making AI more scalable and adaptable for medical applications. In addition, contrastive SSL methods learn generalizable representations that can be used across diverse medical imaging tasks (Khan et al., 2023). This is because contrastive methods often leverage images paired with a textual description in natural language, which provides richer semantic context than the fixed categorical label typically used in supervised image classification (Huang et al., 2023).

However, despite these advancements, translating AI-powered tools into practical implementations warrants careful consideration (Ghassemi et al., 2019). Group bias, characterized by inconsistent model performance across demographic subgroups, can lead to unequal treatment based on, for instance, gender or race (Chen et al., 2019; Glocker et al., 2022). This can potentially exacerbate healthcare disparities (Cullen et al., 2022).

Earlier work has found biases in vision-only models (Seyyed-Kalantari et al., 2020; Glocker et al., 2023; Jones et al., 2023), but fairness of vision-language (VL) models remains understudied. Research into this topic is,

however, hampered by a lack of VL datasets containing detailed demographic information. The study by Luo et al. (2024a) addresses this issue by releasing Harvard-FairVLMed, the first fair VL medical dataset. Using their dataset, the authors evaluate the fairness of two commonly used VL models, CLIP (Radford et al., 2021) and BLIP2 (Li et al., 2023). The authors find that both models exhibit group bias. A new model, FairCLIP, is proposed with a modified loss function that enhances both performance and fairness.

Our contributions can be summarized as follows:

- We replicate the experiments in Luo et al. (2024a) and find that most results are reproducible.
- By evaluating the authors' reasoning, we conclude that the key metric to claim fairness improvements, Equity-Scaled AUC, is flawed. We apply two additional metrics: one from the literature and another as a modification to this key metric. We find that an improvement in fairness of FairCLIP over CLIP is inconsistent.
- We run several experiments in an effort to determine the source of bias. We find no conclusive evidence that the clinical notes, medical pre-training, or group imbalance are fundamental sources of bias.

## 2 Scope of reproducibility

In their paper, Luo et al. (2024a) evaluate performance and fairness of VL models on a glaucoma detection task. Based on the findings of their study, the authors make the following two main claims:

**C1: CLIP exhibits bias, expressed in notable performance disparities between subgroups of an attribute.**

**C2: FairCLIP demonstrates a significant improvement over CLIP in terms of both performance and fairness.**

This study aims to reproduce these claims. Due to computational constraints, we exclude BLIP2 from our reproduction, as the original study primarily benchmarked FairCLIP against CLIP. Furthermore, since CLIP outperformed BLIP2 in the study by Luo et al. (2024a), we focus on evaluating CLIP.

The paper is structured as follows: Section 3 outlines the models, dataset, and experimental setup. Section 4 presents the reproduced results, evaluation of the claims, and our extensions. Finally, Section 5 presents conclusions, limitations, and insights into the reproducibility process.

## 3 Methodology

### 3.1 Model Descriptions

**Contrastive Language-Image Pretraining (CLIP)**: This is a foundational VL model pretrained on a large-scale dataset of image-text pairs collected from the internet, leveraging a contrastive learning objective to align visual and textual representations (Radford et al., 2021). For this study, two Vision Transformers (ViT) are used: 1) ViT-Base with 86 million parameters and a $16 \times 16$ input patch size (B/16), and 2) ViT-Large with 307 million parameters and a $14 \times 14$ input patch size (L/14) (Dosovitskiy et al., 2021).

**FairCLIP**: This model extends CLIP by incorporating the Sinkhorn distance into its loss function during the pre-training phase on the Harvard-FairVLMed dataset. This adjustment aims to minimize the distances between distributions within a demographic group in an attempt to improve fairness. More formally, the goal of improving fairness is defined by:

$$\min_f \sum_\alpha^{\mathcal{A}} d\left(\mathcal{D}_{\{(x_I, x_T, a)\}|f} - \mathcal{D}_{\{(x_I, x_T, a)|a=\alpha\}|f}\right) \tag{1}$$

where $f$ is the pre-trained model (CLIP), $d$ is a distance function (in this case the Sinkhorn distance), $x_I$ and $x_T$ are image and text features, respectively, $a$ is an identity attribute associated with the patient, such

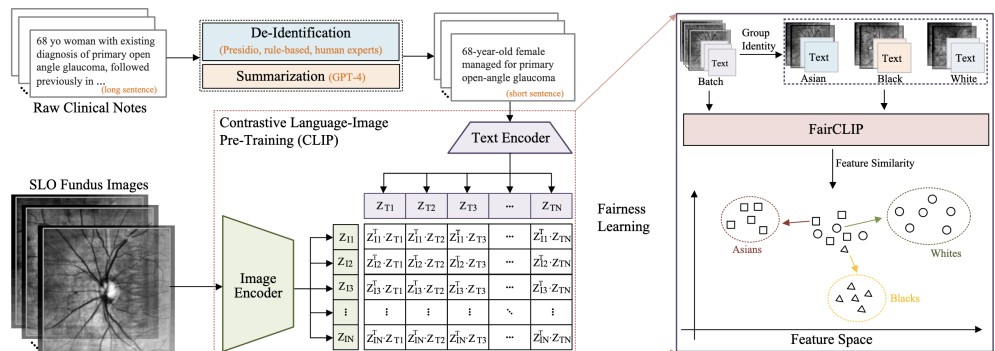

Figure 1: From Luo et al. (2024a). Schematic view of the proposed method, FairCLIP.

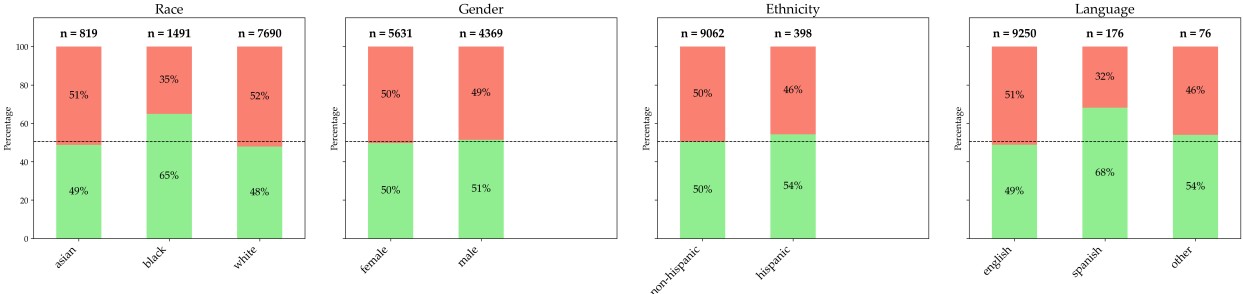

Figure 2: Distribution of positive and negative glaucoma labels across subgroups for each attribute (race, gender, ethnicity, and language). Each bar represents the percentage of positive (green) and negative (red) labels within a subgroup. The dashed horizontal line indicates the overall proportion of positive glaucoma diagnoses in the dataset (50.48%). Sample sizes per subgroup are shown above each bar.

as gender or race, $\mathcal{D}_{\{(x_I, x_T, a)\}|f}$ is the overall data distribution for attribute $a$, and $\mathcal{D}_{\{(x_I, x_T, a)|a=\alpha\}|f}$ is the data distribution for attribute subgroup $\alpha \in \mathcal{A}$. $\mathcal{A}$ can be {Asian, Black, White}, {Female, Male}, {Non-Hispanic, Hispanic}, or {English, Spanish, Others}. Further details about the formulation of the Sinkhorn distance can be found in Section A in the Appendix.

## 3.2 Dataset

The original dataset[1] from Luo et al. (2024a) consists of 10,000 Scanning Laser Ophthalmoscopy (SLO) fundus images paired with a clinical note. The clinical notes provide descriptions of the images along with non-imaging clinical information, including medications, non-imaging test results, and family history. Each image-text pair is associated with six demographic attributes: gender, race, ethnicity, age, preferred language, and marital status. Furthermore, each record has a ground-truth label indicating the presence or absence of glaucoma. The dataset is split into 7000 training, 1000 validation, and 2000 test samples.

Figure 2 visualizes the percentage of positive and negative glaucoma labels for each demographic group. The total number of samples per subgroup is displayed at the top of each bar.

During data processing, Luo et al. (2024a) performed de-identification of the raw clinical notes, where Protective Health Information (PHI), such as name, phone number, and glaucoma diagnosis date, were replaced by placeholders to ensure patient privacy. Furthermore, all de-identified clinical notes were summarized with GPT-4 to be compatible with the token limit of the pre-trained VL models.

---

[1]The Harvard-FairVLMed dataset can be accessed upon request at ophai.hms.harvard.edu/datasets/harvard-fairvlmed10k

### 3.3 Experimental Setup and Code

In this section, we describe the pre-training process of the models and outline the evaluation techniques used. We also provide the exact configurations used to replicate the key tables from Luo et al. (2024a). Our code is available at Github FairCLIP Reproduction.

#### 3.3.1 Pre-trained Models

We employ two pre-trained variants of CLIP. The first one is the natural variant, initialized using the publicly available checkpoints provided by CLIP. The second one is the medical variant, fine-tuned on the Harvard-FairVLMed dataset after being initialized with the official checkpoints. This latter version will be referred to as fine-tuned or FT. During this phase, SLO fundus images are paired with summarized clinical notes to enhance image representation learning. In the code provided by Luo et al. (2024a), model selection during fine-turning was based on the highest AUC validated on the test data across epochs. As this approach constitutes test data leakage, we used the validation set for model selection during training in all experiments presented in this paper.

#### 3.3.2 Evaluation

To evaluate model performance, two approaches are employed. The first is linear probing, which involves training a separate classification model on top of the pre-trained VL model's image representations. Thus the images are used as input, while the clinical summaries are excluded. Specifically, a linear classifier is trained to predict the diagnostic label, following the methodology outlined by He et al. (2021). Additionally, a batch normalization layer is applied before the linear classifier to standardize the feature representations. The second evaluation approach is zero-shot inference, where the model classifies images based on the highest similarity score with text embeddings corresponding to the phrases: "This is glaucoma" and "This is not glaucoma".

#### 3.3.3 Hyperparameters

For this reproducibility study, we followed the hyperparameters documented by Luo et al. (2024a). The fine-tuning of the models runs for 10 epochs. A batch size of 32 was used, and the Adam optimizer (Kingma & Ba, 2014) with a learning rate of 1e-5, $\beta_1$ and $\beta_2$ of 0.1, and weight decay of 6e-5. The fine-tuning of FairCLIP follows the same method and hyperparameter values. For linear probing we used 1000 training epochs. The LARS optimizer (You et al., 2017) was used, with a base learning rate of 0.1, weight decay of 0, and a batch size of 512. For zero-shot evaluation, the same setup as CLIP pre-training was used.

#### 3.3.4 Metrics

The models were evaluated on the classification task with a focus on fairness, performance, and the performance-fairness trade-off. The following four metrics were introduced in the original paper. Additionally, we introduce two fairness metrics to expand on the fairness evaluation.

**Demographic Parity Difference (DPD)** (Dwork et al., 2012; Agarwal et al., 2018): This metric measures the extent to which the model achieves statistical parity. In the context of this paper, statistical parity is satisfied when the probability of receiving a positive glaucoma diagnosis is equal across all demographic groups within the same attribute. A DPD greater than zero suggests a disparity in diagnosis rates across groups, indicating potential bias. Formally, DPD is defined as the maximum difference in positive prediction rates between any two groups:

$$\text{DPD} = \max_{i,j \in \{1,\dots,n\}} \left| \mathbb{P}(\hat{Y} = 1 \mid A = a_i) - \mathbb{P}(\hat{Y} = 1 \mid A = a_j) \right| \qquad (2)$$

where $\hat{Y}$ is the predicted label and $A$ is the protected attribute with $n$ subgroups $\{a_1, \dots, a_n\}$.

**Difference in Equalized Odds (DEOdds)** (Agarwal et al., 2018; Hardt et al., 2016): The DEOdds measures the disparities in True Positive Rate (TPR) and False Positive Rate (FPR) between demographic

groups. In this paper, a DEOdds of zero indicates that the model predicts glaucoma and non-glaucoma cases with equal accuracy and error rates across all demographic subgroups of an attribute, ensuring fairness in both correct and incorrect classifications. It is computed as:

$$\text{DEOdds} = \max_{i,j \in \{1,\ldots,n\}} \left( \left| \text{TPR}_{a_i} - \text{TPR}_{a_j} \right| + \left| \text{FPR}_{a_i} - \text{FPR}_{a_j} \right| \right) \tag{3}$$

where $\text{TPR}_{a_i} = \text{TP}/(\text{TP} + \text{FN})$ and $\text{FPR}_{a_i} = \text{FP}/(\text{FP} + \text{TN})$.

**Area under the Receiver Operating Characteristic Curve (AUC)** (Bradley, 1997): In this paper, the sole performance metric used is AUC. AUC measures the model's ability to classify correctly across all possible thresholds, making it more robust than accuracy, which depends on a single threshold. An AUC of 50% reflects random guessing, while an AUC of 100% indicates perfect model performance, regardless of the threshold chosen.

**ES-AUC** (Luo et al., 2024b): The Equity-Scaled AUC (ES-AUC) metric is designed to assess both model performance and fairness in a unified manner. It extends AUC by incorporating group-wise disparity as fairness consideration. Specifically, ES-AUC penalizes performance disparities by scaling the overall AUC with the absolute sum of discrepancies between the global AUC and each group-wise AUC:

$$\text{ES-AUC} = \frac{\text{AUC}}{1 + \sum_{i=1}^{n} |\text{AUC} - \text{AUC}_{a_i}|} \tag{4}$$

where $A$ is the protected attribute with $n$ subgroups $\{a_1, \ldots, a_n\}$, and $\text{AUC}_{a_i}$ is the AUC calculated on samples from group $a_i$.

**Precision Disparity (PD)** (Castelnovo et al., 2022): In this study, we add the Precision Disparity (PD) as a fairness metric. PD measures disparities in precision, or Positive Predictive Value (PPV), across demographic groups. A score of zero indicates that the precision is completely equal across all demographic groups. In the context of this task, a low PD score means that individuals from all groups have the same likelihood of a positive glaucoma diagnosis being correct. This is crucial in healthcare settings, where fairness in the reliability of diagnoses directly impacts trust and treatment outcomes. PPV and PD are formally defined as:

$$\text{PD} = \max_{i,j \in \{1,\ldots,n\}} \left| \text{PPV}_{a_i} - \text{PPV}_{a_j} \right|, \text{ with: } \text{PPV}_{a_i} = \frac{\text{TP}_{a_i}}{\text{TP}_{a_i} + \text{FP}_{a_i}} \tag{5}$$

where $\text{TP}_{a_i}$ and $\text{FP}_{a_i}$ denote the number of true positives and false positives, respectively, for subgroup $a_i$ of the protected attribute $A$.

Unlike DPD, which enforces statistical parity regardless of disease prevalence, PD ensures fairness by maintaining consistent prediction accuracy. This distinction is particularly important in medical diagnosis, where prevalence differences are clinically meaningful, and forcing equal diagnosis rates could lead to misdiagnoses or suboptimal treatment decisions.

Additionally, PD highlights a different aspect of fairness than DEOdds. While DEOdds captures disparities in detection ability (TPR) and false alarms (FPR), it does not measure whether positive predictions themselves are equally likely to be correct (i.e., PPV). This reliability becomes important in clinical scenarios where disease prevalence differs by demographic group. If glaucoma is more prevalent in one group, even with equal TPR and FPR, the model's positive predictions will be more accurate for that group. Therefore, where DEOdds measures if the model treats different groups equally, PD measures if the model's positive predictions are equally trustworthy. As further illustrated in Section 4.2, glaucoma prevalence varies across demographic groups. Therefore, combining PD with DEOdds provides a comprehensive fairness assessment that considers both prediction distributions and predictive reliability, ensuring that fairness assessments align with both ethical and clinical considerations.

**Mean Absolute Deviation**: We use the Mean Absolute Deviation (MAD) as a fairness metric to complement ES-AUC. As will be further discussed in more detail in Section 4.2, the ES-AUC metric alone cannot be used to make definitive claims about fairness as it primarily evaluates the trade-off between performance and fairness. Consequently, a larger model may increase ES-AUC without actually improving fairness. To address this limitation, we decompose ES-AUC into a performance component (the overall AUC), and a fairness component, which is defined as:

$$MAD = \frac{1}{|\mathcal{A}|} \sum_{a \in \mathcal{A}}^{\mathcal{A}} |AUC - AUC_a| \tag{6}$$

where we normalize by the number of demographic subgroups $|\mathcal{A}|$ to ensure that the fairness measurement is independent of the number of subgroups. MAD explicitly quantifies fairness by computing the average absolute deviation in group-wise AUC values. A lower MAD value indicates greater fairness, while a higher MAD suggests greater disparity in model performance across groups, and thus lower fairness.

Note that the MAD metric is not intended as a replacement for ES-AUC but rather as a complementary measure. By presenting ES-AUC alongside MAD, we enable a more accurate evaluation of fairness improvements in a model. ES-AUC retains its role in assessing the overall performance-fairness trade-off, while MAD provides a direct and interpretable fairness metric. This ensures that fairness claims can be properly substantiated rather than inferred solely from ES-AUC changes.

Other fairness metrics suitable here have been used in the literature. True Positive Rate (TPR) disparity is used in Seyyed-Kalantari et al. (2020), while the fairness gap (the difference in AUC between the best- and worst-performing subgroup) is used in Khan et al. (2023). However, we found it valuable to stay close to the definition of fairness in Luo et al. (2024a). ES-AUC was motivated as a balance between performance and fairness, and it is the key evaluation metric for many of the experiments in the original paper. Thus, we decompose ES-AUC into its two parts, using one part (MAD) as a fairness metric.

### 3.3.5 Experiments: Finding the Source of Bias

To investigate the source of bias, we conducted additional experiments. We evaluate the change in model bias after fine-tuning the model with the clinical notes, pre-training the model on the medical domain, and we consider the effect of group imbalance. We also evaluate whether the model can predict the demographic attribute gender from the image. If the model lacks this ability, the gender bias does not originate from the images.

**Impact of Clinical Notes**: We research the impact of the summarized clinical notes on fairness as a potential source of bias. CLIP (L/14) was fine-tuned on the images paired with the text "This is glaucoma" or "This is not glaucoma", for a positive and negative example, respectively. In the original paper, the authors tested the difference between this setup and the baseline of fine-tuning with the clinical note via linear probing. This experiment in Luo et al. (2024a) was used to determine the effect of vision-only versus multimodal features; we use it to analyze the source of the bias.

**Medical Pre-training**: In Khan et al. (2023) the authors find that medical pre-training improves overall performance but consistently degrades fairness. Building on this finding, we investigate whether medical pre-training contributes to bias by comparing the fairness of CLIP and CLIP-FT. If a disparity emerges, it would suggest that medical pre-training is a source of bias.

**Group Imbalance**: Larrazabal et al. (2020) found that an unequal gender distribution induces bias, resulting in poorer performance for the minority group. Building on this finding, we investigate whether group imbalance contributes to bias by comparing the performance between the minority and majority groups.

**Predicting Attributes from Images**: We focused exclusively on predicting gender from the image. Gender references appear in nearly half of the 10,000 clinical notes, much more than for the other attributes (see Table 7 in the Appendix.) This allows the model capture potential patterns in fundus images specific to a gender and associate them to the corresponding gender. For this experiment we use the CLIP-FT L/14 model in a zero-shot setting. During inference, we compute the similarity between the encoded image and

| Attribute | Model | DPD ↓ | DEOdds ↓ | AUC ↑ | ES-AUC ↑ | Group-wise AUC ↑ | | |
|---|---|---|---|---|---|---|---|---|
| | | | | | | **Asian** | **Black** | **White** |
| Race | CLIP | $5.43 \pm 0.44$ | $14.43 \pm 1.98$ | $77.21 \pm 0.05$ | $72.46 \pm 0.24$ | $79.88 \pm 0.3$ | $73.81 \pm 0.2$ | $77.71 \pm 0.05$ |
| | CLIP-FT | $\mathbf{3.75} \pm 1.18$ | $\mathbf{8.85} \pm 0.91$ | $\mathbf{78.15} \pm 2.99$ | $\mathbf{73.77} \pm 2.36$ | $\mathbf{80.81} \pm 3.27$ | $\mathbf{75.53} \pm 2.07$ | $\mathbf{78.8} \pm 2.95$ |
| | | | | | | **Female** | **Male** | |
| Gender | CLIP | $\mathbf{0.35} \pm 0.17$ | $\mathbf{5.54} \pm 0.06$ | $77.21 \pm 0.05$ | $72.45 \pm 0.03$ | $74.22 \pm 0.04$ | $80.79 \pm 0.07$ | |
| | CLIP-FT | $1.67 \pm 1.44$ | $6.7 \pm 0.4$ | $\mathbf{78.15} \pm 2.99$ | $\mathbf{73.72} \pm 2.72$ | $\mathbf{75.41} \pm 2.9$ | $\mathbf{81.41} \pm 3.06$ | |
| | | | | | | **Non-Hispanic** | **Hispanic** | |
| Ethnicity | CLIP | $17.71 \pm 0.76$ | $\mathbf{17.17} \pm 0.75$ | $77.21 \pm 0.05$ | $71.86 \pm 0.18$ | $77.44 \pm 0.05$ | $69.99 \pm 0.24$ | |
| | CLIP-FT | $\mathbf{13.53} \pm 3.55$ | $19.3 \pm 1.74$ | $\mathbf{78.15} \pm 2.99$ | $\mathbf{74.03} \pm 4.09$ | $\mathbf{78.43} \pm 2.92$ | $\mathbf{72.75} \pm 4.78$ | |
| | | | | | | **English** | **Spanish** | **Others** |
| Language | CLIP | $\mathbf{14.68} \pm 1.4$ | $33.63 \pm 0.33$ | $77.21 \pm 0.05$ | $\mathbf{70.62} \pm 0.17$ | $77.18 \pm 0.05$ | $\mathbf{84.09} \pm 0.0$ | $\mathbf{74.79} \pm 0.21$ |
| | CLIP-FT | $15.62 \pm 3.85$ | $\mathbf{21.67} \pm 12.5$ | $\mathbf{78.15} \pm 2.99$ | $69.54 \pm 2.27$ | $\mathbf{78.56} \pm 3.08$ | $80.3 \pm 6.33$ | $68.88 \pm 2.18$ |

Table 1: Linear probing results of CLIP and CLIP-FT, both for architecture L/14.

the encodings of the two prompts: "This is a fundus photo of a female" and "This is a fundus photo of a male."

### 3.4 Computational Resources and Environmental Impact

All experiments ran on NVIDIA A100 GPUs (40GB). Replication is recommended on A100s or equivalent hardware. Total compute cost, including pre-training, fine-tuning, and evaluation, was ∼16,039 SBUs. Linear probing averaged eight hours per model; zero-shot evaluation took ∼20 minutes per model.

Estimated emissions were 7.35 kgCO$_2$eq (0% offset), calculated using the MachineLearning Impact calculator with a carbon efficiency of 0.26 kg/kWh (Dutch grid; (CBS, 2023)).

## 4 Results

In the following section, we first show the reproduction of the two tables that form the backbone of the analysis in Luo et al. (2024a). Next, we dissect main claims **C1** and **C2** into more precise subclaims for evaluation. We then discuss whether the main claims can be derived from these subclaims. Additionally, we present the results of the extended experiments. All numerical results are presented as the mean and standard deviation across three seeds.

### 4.1 Reproducibility Results

Table 1 presents the results for CLIP (L/14) evaluated with linear probing. The table shows the performance of CLIP and CLIP-FT across four different protected attributes. The results are generally similar to the findings of Luo et al. (2024a). Furthermore, in Table 2 we present the results for CLIP and FairCLIP (both fine-tuned) evaluated in the zero-shot setting. The table shows the performance of both architectures (B/16 and L/14) across the four attributes. The results are, again, similar to the results of the original paper, showing that their results are reproducible.

#### 4.1.1 Subclaims of C1: Bias of CLIP

**C1a: CLIP exhibits disparities between group-wise AUC values in the linear probing setting.** We are able to reproduce this claim. Based on the results in Table 1, we indeed see disparities in group-wise AUC values. This indicates that CLIP exhibits bias in the linear probing setting. This finding is significant. For an overview of p-values for CLIP, see Table 9 in the Appendix.

**C1b: In the linear probing setting, the Asian, male, non-Hispanic, and Spanish subgroups are the best performing in their respective attribute groups.** We are also able to reproduce this claim. Table 1 shows that the Asian, male, non-Hispanic and Spanish subgroups are the best performing in their respective attribute groups. This is derived by taking the mean over the two models per attribute.

**C1c: CLIP exhibits disparities between group-wise AUC values in the zero-shot setting.** This claim was also reproducible. Table 2 shows that CLIP exhibits bias in the zero-shot setting, since there is a disparity in group-wise AUC values. This finding was also significant. For an overview of p-values of the significance of the disparity in group-wise AUC, see Table 10 in the Appendix.

**C1d: In the zero-shot setting, the Asian, male, non-Hispanic, and English subgroups are the best performing in their respective attribute groups.** We were also able to reproduce this claim. In Table 2 it is visible that Asian, male, non-Hispanic and English are the best-performing subgroups of their attribute group.

**C1e: Medical VL pre-training enhances the performance-fairness tradeoff, expressed in a higher ES-AUC, across all attributes except language.** Finally, we were also able to reproduce this claim. As shown in Table 1, medical VL pre-training achieves a higher ES-AUC across all attributes except language. However, this finding is not significant, meaning that this claim is inconclusive. For an overview of p-values testing the significance of the difference between CLIP and CLIP-FT, see Table 11 in the Appendix.

### 4.1.2 Subclaims for C2: FairCLIP Improves Performance and Fairness

Here we deviate from the reasoning used in Luo et al. (2024a). In the original paper, the authors compare CLIP and FairCLIP across two different architectures (B/16 and L/14), resulting in a total of four models. In the table presenting the scores for these models, the best-performing model for each metric and attribute is highlighted in bold, suggesting that all four models are compared simultaneously. To ensure a fair evaluation, we will compare FairCLIP and CLIP within each architecture for each attribute separately.

**C2a: FairCLIP improves DPD relative to CLIP.** Although we were largely able to reproduce the results related to this claim, we cannot fully validate it. In Table 2 it is visible that FairCLIP outperforms CLIP on DPD for both architectures only for the ethnicity attribute. For race and gender, FairCLIP is better for B/16, but not for L/14. For language it is the other way around. Because of this inconsistency, we cannot validate the claim.

**C2b: FairCLIP improves DEOdds relative to CLIP.** Although we reproduced most results for this claim, we again cannot fully validate it. Only for the ethnicity attribute does FairCLIP outperform CLIP in terms of DEOdds for both architectures. For the three other attributes, FairCLIP is better for B/16, but not for L/14. Because of this inconsistency, we cannot validate the claim.

**C2c: FairCLIP improves ES-AUC relative to CLIP.** In this case, we were not able to reproduce the results supporting this claim. In the table in the original paper, FairCLIP has the best ES-AUC for all attributes. In our case, as we show in Table 2, FairCLIP does not outperform CLIP on either of the two architectures for gender and language. For race, FairCLIP is better only for B/16, for ethnicity it is better only for L/14. Thus, we cannot validate this claim.

**C2d: FairCLIP improves AUC relative to CLIP.** For the final claim we were also unable to reproduce the results supporting it. In the original paper, FairCLIP has the best AUC for all attributes. Under our reproducibility, as is visible in Table 2, FairCLIP does not outperform CLIP in terms of AUC on either architecture for the gender attribute. For the other three attributes, FairCLIP outperforms CLIP only for the B/16 architecture. We thus cannot validate this claim.

In addition to finding no consistent improvements of FairCLIP relative to CLIP, we also found that all cases of improvement were not significant. See Table 12 in the Appendix for an overview of p-values comparing CLIP to FairCLIP.

### 4.2 Evaluation of Main Claims

We have presented the results of replicating the experiments in Luo et al. (2024a). Our findings largely align with the trends observed in the original paper. Based on the results, we were able to validate claims **C1a-C1d** (**C1e** is inconclusive), but we were unable to validate claims **C2a-C2d**. In this section, we examine

| Attribute | Model | DPD ↓ | DEOdds ↓ | AUC ↑ | ES-AUC ↑ | Group-wise AUC ↑ | | |
|---|---|---|---|---|---|---|---|---|
| | | | | | | **Asian** | **Black** | **White** |
| Race | CLIP B/16 | $16.18 \pm 4.24$ | $17.78 \pm 2.78$ | $67.89 \pm 3.65$ | $60.84 \pm 2.23$ | **72.46** $\pm 6.22$ | **72.64** $\pm 3.37$ | $65.68 \pm 3.87$ |
| | FairCLIP | **1.40** $\pm 0.74$ | **4.68** $\pm 3.51$ | **68.59** $\pm 1.74$ | **63.53** $\pm 2.88$ | $71.71 \pm 2.65$ | $72.03 \pm 3.54$ | **67.06** $\pm 1.84$ |
| | CLIP L/14 | **11.00** $\pm 6.00$ | **11.94** $\pm 6.44$ | **69.04** $\pm 2.32$ | **65.62** $\pm 3.19$ | **72.59** $\pm 2.45$ | **69.87** $\pm 1.98$ | **68.17** $\pm 2.89$ |
| | FairCLIP | $22.62 \pm 5.47$ | $26.03 \pm 8.29$ | $64.10 \pm 6.21$ | $59.98 \pm 4.91$ | $66.86 \pm 6.74$ | $65.80 \pm 8.24$ | $62.61 \pm 6.32$ |
| | | | | | | **Female** | **Male** | |
| Gender | CLIP B/16 | $5.01 \pm 2.58$ | $8.96 \pm 2.20$ | **67.89** $\pm 3.65$ | **63.89** $\pm 2.93$ | **65.20** $\pm 3.25$ | **71.44** $\pm 4.13$ | – |
| | FairCLIP | **3.01** $\pm 1.84$ | **6.15** $\pm 2.73$ | $67.05 \pm 0.57$ | $62.88 \pm 0.05$ | $64.15 \pm 0.22$ | $70.80 \pm 1.06$ | – |
| | CLIP L/14 | **2.65** $\pm 2.02$ | **6.24** $\pm 1.67$ | **69.04** $\pm 2.32$ | **64.95** $\pm 2.12$ | **66.27** $\pm 2.23$ | **72.56** $\pm 2.40$ | – |
| | FairCLIP | $7.40 \pm 2.71$ | $8.78 \pm 4.71$ | $63.35 \pm 3.86$ | $60.49 \pm 4.77$ | $61.29 \pm 4.57$ | $66.11 \pm 2.89$ | – |
| | | | | | | **Non-Hispanic** | **Hispanic** | |
| Ethnicity | CLIP B/16 | $13.11 \pm 4.40$ | $16.80 \pm 8.78$ | $67.89 \pm 3.65$ | **62.77** $\pm 1.03$ | $68.17 \pm 3.81$ | **60.07** $\pm 0.27$ | – |
| | FairCLIP | **8.09** $\pm 1.90$ | **13.48** $\pm 2.07$ | **68.32** $\pm 1.13$ | $62.35 \pm 0.36$ | **68.68** $\pm 1.17$ | $59.11 \pm 0.39$ | – |
| | CLIP L/14 | $10.78 \pm 5.88$ | $14.32 \pm 5.98$ | $69.04 \pm 2.32$ | $62.44 \pm 3.21$ | **69.43** $\pm 2.38$ | $58.73 \pm 5.03$ | – |
| | FairCLIP | **6.96** $\pm 4.24$ | **9.33** $\pm 6.05$ | **69.14** $\pm 4.27$ | **66.07** $\pm 6.73$ | $69.31 \pm 4.10$ | **64.37** $\pm 8.37$ | – |
| | | | | | | **English** | **Spanish** | **Others** |
| Language | CLIP B/16 | **12.00** $\pm 2.59$ | $20.86 \pm 7.60$ | $67.89 \pm 3.65$ | **60.12** $\pm 1.61$ | $67.98 \pm 3.85$ | **62.41** $\pm 3.53$ | **60.54** $\pm 0.52$ |
| | FairCLIP | $12.67 \pm 3.90$ | **18.60** $\pm 1.91$ | **68.38** $\pm 2.67$ | $54.56 \pm 2.92$ | **68.69** $\pm 2.65$ | $53.13 \pm 5.60$ | $58.55 \pm 1.94$ |
| | CLIP L/14 | $14.15 \pm 1.85$ | **15.89** $\pm 7.11$ | **69.04** $\pm 2.32$ | **62.99** $\pm 2.20$ | **69.08** $\pm 2.57$ | **65.53** $\pm 2.58$ | $63.13 \pm 3.52$ |
| | FairCLIP | **10.23** $\pm 2.34$ | $16.49 \pm 2.45$ | $66.52 \pm 3.23$ | $59.06 \pm 3.81$ | $66.50 \pm 3.49$ | $60.32 \pm 6.45$ | $59.74 \pm 1.43$ |

Table 2: Zero-shot transfer results of CLIP and FairCLIP (both fine-tuned). FairCLIP uses the same architecture as CLIP right above it.

whether the metrics in the original paper are relevant to assess fairness and, based on this, if the main claims **C1** and **C2** can be derived from the subclaims.

**DPD as Fairness Metric:** In Luo et al. (2024a) it is argued that DPD and DEOdds are widely used in fairness evaluation to determine whether certain groups are systematically disadvantaged. As discussed in Section 3.3.4, DPD measures the extent to which the model predicts the same probability of a positive outcome for all subgroups in a demographic group. Thus, for this metric to effectively assess fairness, the probability of a positive outcome must be the same for all subgroups in the population. However, empirical evidence contradicts this assumption. A study by Rudnicka et al. (2006) found that men are 1.37 times more likely to develop open-angle glaucoma (the most common form of glaucoma) compared to women. Additionally, it was found that Black individuals face a significantly higher risk of developing open-angle glaucoma compared to White and Asian individuals. Thus, as glaucoma prevalence varies across demographic groups, the probability of a positive diagnosis is unequal across groups. Consequently, DPD may not be a suitable fairness metric in this study, as it assumes an equality that does not naturally exist.

**ES-AUC as Fairness Metric:** The authors explicitly use ES-AUC as a key measure of fairness, as highlighted in claim **C2c**. Throughout the paper, ES-AUC is repeatedly referenced to support fairness claims, suggesting that improvements in ES-AUC imply greater equity across demographic groups. However, ES-AUC was introduced as a metric to evaluate the trade-off between performance and fairness, rather than fairness itself. While it accounts for disparities among subgroups, it does so by penalizing overall AUC when subgroup disparities are large, rather than enforcing a strict trade-off between performance and fairness. This means that an increase in ES-AUC does not necessarily imply an improvement in fairness, as it does not directly measure bias reduction. To illustrate this, consider a scenario where a model's overall performance (i.e., AUC) improves, while the disparity among subgroups remains unchanged. For instance, if all subgroup AUC values increase uniformly, ES-AUC will rise solely due to the overall performance gain, even though fairness has not improved. More importantly, if the disparity between subgroups increases, but the overall AUC improves, the ES-AUC may still increase despite worsening fairness. For example, suppose a model initially has an overall AUC of 60% and has no disparity among its subgroups. Therefore, the ES-AUC would simply be equal to the overall AUC. If the overall AUC increases to 80%, with group-wise AUC values of 90%, 85%, and 65%, this would give:

$$\text{ES-AUC} = \frac{80}{1 + (0.10 + 0.05 + 0.15)} = \frac{80}{1.30} \approx 61.54\%,$$

| Attribute | Model | PD ↓ | MAD ↓ |
|---|---|---|---|
| Race | CLIP | 14.43 ± 1.88 | 2.19 ± 0.17 |
| | CLIP FT | 9.47 ± 0.14 | 1.98 ± 0.59 |
| Gender | CLIP | 5.70 ± 0.29 | 3.29 ± 0.03 |
| | CLIP FT | 4.37 ± 3.06 | 3.00 ± 0.25 |
| Ethnicity | CLIP | 17.64 ± 0.19 | 3.73 ± 0.14 |
| | CLIP FT | 19.26 ± 2.18 | 2.84 ± 1.15 |
| Language | CLIP | 9.62 ± 2.50 | 3.11 ± 0.08 |
| | CLIP FT | 14.44 ± 3.61 | 4.15 ± 1.97 |

Table 3: Linear probing results of CLIP and CLIP-FT (both L/14) for the PD and MAD metrics.

where we first convert percentages to decimal form and compute their absolute differences (90%−80%=0.10, etc.) in the denominator. Although the disparity increased significantly, the ES-AUC still increased due to an increase in performance. Thus, while ES-AUC penalizes disparities in group-wise AUC values, it should not be used in isolation to claim fairness improvements.

### 4.2.1 Validation of Main Claims

Based on the subclaims discussed in Section 4.1.1, the authors conclude that CLIP exhibits bias, forming the foundation of main claim **C1**. This claim logically follows from its subclaims and it does not rely on either ES-AUC or DPD. In addition, we successfully reproduced claims **C1a**-**C1d**. Thus, our results support the validity of claim **C1**.

All subclaims for claim **C2** were invalidated. In addition, as we have shown, the use of DPD for this task is inappropriate, and ES-AUC was misinterpreted as fairness metric. Thus, we cannot validate claim **C2**. In the next section, we will expand the fairness analysis of FairCLIP, providing a more comprehensive evaluation.

### 4.3 Results of Extensions: PD and MAD

Table **??** presents the results for CLIP and CLIP-FT on the new metrics, maintaining the same configurations as Table 1. The MAD metric reveals substantial disparities in group-wise AUCs, reinforcing the claim that CLIP exhibits bias.

Table 4 presents the results for CLIP and FairCLIP on the new metrics using the same configurations as Table 2. For the B/16 architecture, FairCLIP achieves a lower PD than CLIP in three out of four protected attributes, suggesting some improvement in prediction disparity. However, for the L/14 architecture, FairCLIP outperforms CLIP in only one attribute, indicating inconsistent gains.

When evaluating the MAD metric, CLIP B/16 consistently outperforms FairCLIP B/16 across all attributes except race, indicating that FairCLIP does not consistently mitigate performance disparities across all attributes. Similarly, for the L/14 architecture. CLIP achieves the lowest MAD for race and language, whereas FairCLIP performs better for gender and language, showing no clear advantage across the attributes.

Considering both architectures and attributes, we find no significant overall improvement in fairness for FairCLIP. These extended results, therefore, do not support the claim that FairCLIP improves fairness compared to CLIP.

### 4.4 Results of Experiments: Finding the Source of Bias

**Impact of Clinical Notes**: Table 5 shows the results of fine-tuning on only the image (V) and the image paired with the summarized clinical note (VL). Looking at the MAD column, it is visible that VL fine-tuning exhibits less bias than vision-only fine-tuning. However, the values are close together for each attribute and overlap when taking the standard deviation into account. In fact, for all other metrics the values are also close together. This indicates that vision-only or VL fine-tuning is not a fundamental source of bias in CLIP.

**Medical Pre-training**: Considering Table 1, looking at DEOdds, there is no clear improvement or degradation of CLIP compared to CLIP-FT. This suggests that medical pre-training is not a key source of bias.

| Attribute | Model | PD ↓ | MAD ↓ |
|---|---|---|---|
| Race | CLIP (ViT-B/16) | 17.78 ± 2.48 | 3.84 ± 0.71 |
|  | FairCLIP (ViT-B/16) | **4.68** ± 3.51 | **2.70** ± 1.47 |
|  | CLIP (ViT-L/14) | **10.91** ± 7.32 | **1.75** ± 0.72 |
|  | FairCLIP (ViT-L/14) | 21.96 ± 7.81 | 2.27 ± 1.22 |
| Gender | CLIP (ViT-B/16) | 8.96 ± 2.20 | **3.12** ± 0.50 |
|  | FairCLIP (ViT-B/16) | **6.15** ± 2.73 | 3.32 ± 0.42 |
|  | CLIP (ViT-L/14) | **6.24** ± 1.67 | 3.14 ± 0.37 |
|  | FairCLIP (ViT-L/14) | 8.69 ± 4.78 | **2.41** ± 1.03 |
| Ethnicity | CLIP (ViT-B/16) | 16.80 ± 8.78 | **4.05** ± 2.03 |
|  | FairCLIP (ViT-B/16) | **13.48** ± 1.40 | 4.79 ± 0.68 |
|  | CLIP (ViT-L/14) | 14.32 ± 5.98 | 5.35 ± 2.67 |
|  | FairCLIP (ViT-L/14) | **9.30** ± 1.71 | **2.47** ± 2.16 |
| Language | CLIP (ViT-B/16) | **4.53** ± 2.64 | **4.33** ± 2.45 |
|  | FairCLIP (ViT-B/16) | 8.93 ± 7.21 | 8.47 ± 0.77 |
|  | CLIP (ViT-L/14) | **9.13** ± 2.40 | **3.26** ± 2.53 |
|  | FairCLIP (ViT-L/14) | 9.20 ± 1.41 | 4.39 ± 4.23 |

Table 4: Zero-shot transfer results of CLIP and FairCLIP (both FT) for the PD and MAD metrics.

| Attribute | Model | DPD ↓ | DEOdds ↓ | AUC ↑ | ES-AUC ↑ | PD ↓ | MAD ↓ | Group-wise AUC ↑ | | |
|---|---|---|---|---|---|---|---|---|---|---|
|  |  |  |  |  |  |  |  | Asian | Black | White |
| Race | CLIP-V | 5.01 ± 1.37 | 12.74 ± 2.54 | **81.70** ± 0.19 | **75.34** ± 1.93 | 12.74 ± 2.54 | 2.83 ± 0.84 | **84.82** ± 1.51 | **77.15** ± 1.63 | **82.53** ± 0.46 |
|  | CLIP-VL | **3.75** ± 1.18 | **8.85** ± 0.91 | 78.15 ± 2.99 | 73.77 ± 2.36 | **9.47** ± 0.14 | **1.98** ± 0.59 | 80.81 ± 3.27 | 75.53 ± 2.07 | 78.8 ± 2.95 |
|  |  |  |  |  |  |  |  | Female | Male | |
| Gender | CLIP-V | **0.92** ± 0.98 | **7.39** ± 1.19 | **81.70** ± 0.19 | **77.05** ± 0.73 | 7.39 ± 1.19 | 3.02 ± 0.39 | **78.91** ± 0.57 | **84.95** ± 0.22 | |
|  | CLIP-VL | 1.67 ± 1.44 | 6.7 ± 0.4 | 78.15 ± 2.99 | 73.72 ± 2.72 | **4.37** ± 3.06 | **3.00** ± 0.25 | 75.41 ± 2.9 | 81.41 ± 3.06 | |
|  |  |  |  |  |  |  |  | Non-Hispanic | Hispanic | |
| Ethnicity | CLIP-V | **10.11** ± 1.43 | **13.18** ± 4.20 | **81.70** ± 0.19 | **76.49** ± 0.69 | **13.18** ± 4.20 | 3.11 ± 0.58 | **81.96** ± 0.22 | **75.14** ± 0.96 | |
|  | CLIP-VL | 13.53 ± 3.55 | 19.3 ± 1.74 | 78.15 ± 2.99 | 74.03 ± 4.09 | 19.26 ± 2.18 | **2.84** ± 1.15 | 78.43 ± 2.92 | 72.75 ± 4.78 | |
|  |  |  |  |  |  |  |  | English | Spanish | Others |
| Language | CLIP-V | 17.20 ± 0.82 | 32.07 ± 4.71 | **81.70** ± 0.19 | **71.65** ± 0.60 | **10.72** ± 6.15 | 4.68 ± 0.34 | **82.16** ± 0.23 | **83.33** ± 3.13 | **70.96** ± 1.35 |
|  | CLIP-VL | **15.62** ± 3.85 | **21.67** ± 12.5 | 78.15 ± 2.99 | 69.54 ± 2.27 | 14.44 ± 3.61 | **4.15** ± 1.97 | 78.56 ± 3.08 | 80.3 ± 6.33 | 68.88 ± 2.18 |

Table 5: Linear probing results of CLIP L/14, fine-tuned on images only (V) or images and summaries (VL).

**Group Imbalance**: Similar to the findings of Luo et al. (2024a), Table 1 shows that for three out of four attributes the minority group has the best performance. This indicates that group imbalance is not a key source of bias in CLIP.

**Predicting Attributes from Images**: In Table 6 the results of predicting gender are presented, where the glaucoma prediction results is a baseline. Note that an AUC of 50% represents a performance similar to a random prediction. Although the observed gender AUC is substantially lower than the glaucoma AUC and is relatively close to 50%, we cannot reject the possibility that the model is able to predict the attribute gender. This suggests the bias might originate from the images.

In summary, we find no conclusive evidence that fine-tuning with the summarized clinical notes, pre-training on the medical domain, or the group imbalance are fundamental sources of bias. Furthermore, we found the model may have the ability to predict gender from fundus images.

## 5 Discussion

We largely reproduced the results of Luo et al. (2024a). Our findings confirm the claim that CLIP exhibits bias in glaucoma detection. However, despite FairCLIP being presented as a fairness-improving method, our extended results do not provide strong evidence that FairCLIP improves fairness compared to CLIP. While there are cases where FairCLIP reduces group-wise disparities, the improvements are inconsistent across architectures and attributes. This challenges the assumption that FairCLIP universally enhances fairness and suggests that fairness gains might be architecture- or attribute-dependent, rather than a generalizable improvement. Furthermore, we showed that the observed bias does not stem from the summarized clinical notes, medical pre-training or group imbalance. Finally, we cannot rule out that the bias does not originate from the images, as supported by CLIP's modest ability to predict the gender attribute.

| Prediction | Accuracy | AUC ↑ |
|---|---|---|
| Glaucoma | 61.96 ± 2.18 | 69.04 ± 2.32 |
| Gender | 50.10 ± 2.24 | 55.40 ± 1.54 |

Table 6: Zero-shot evaluation of CLIP (ViT-L/14) for predicting gender.

## 5.1 Limitations

Our study has several limitations that should be considered. First, there is a lack of established benchmarks for evaluating the bias of CLIP in medical settings, making it challenging to contextualize our findings against a broader standard. Furthermore, our experiments focus specifically on glaucoma prediction, which may limit the generalizability of the findings to other medical or non-medical applications. Lastly, due to limited computational resources, we did not explore larger-scale experiments, such as fine-tuning on a different dataset or evaluating additional vision-language models beyond CLIP and FairCLIP. This may limit the depth of our analysis and the robustness of our conclusions.

## 5.2 Future Research

Understanding the origins of bias in CLIP remains an open challenge. First, we suggest future search can focus on increasing understanding about the predictions of the model. Although our gender prediction experiment offers an initial insight, similar investigations should be conducted for other attributes, both in the vision-language and vision-only setting. While the ability to distinguish between groups is not inherently problematic, it may contribute to the bias. Furthermore, future research should aim to identify the source of bias of CLIP. For instance, different demographic groups may facilitate easier glaucoma detection. Another source of bias can be that CLIP encountered annotated glaucoma images during pre-training. We leave exploring both options for future research.

Additionally, investigating mitigation strategies beyond FairCLIP could provide further insights into reducing bias in medical vision-language models. For example, adversarial debiasing techniques use a secondary model to predict protected attributes, encouraging the main model to focus on task-relevant features while ignoring sensitive information. Furthermore, domain adaptation mitigates bias by aligning feature distributions across domains, improving generalization and reducing domain-specific biases. Finally, while FairCLIP employed the Sinkhorn distance to create a fairness-aware loss function, other fairness-aware loss functions remain relatively underexplored. These approaches could provide valuable insights into reducing bias in medical vision-language models.

## 5.3 Reproducibility Analysis

**What was easy**: The complete codebase was publicly available. This allowed us to access and execute the experiments without significant barriers. Additionally, the original paper mostly provided a clear and comprehensive explanation of FairCLIP, making it relatively easy to understand and implement.

**What was difficult**: Despite the helpful resources, we encountered several challenges. Firstly, there were some ambiguities in the text. For instance, the terms fine-tuning and pre-training were used interchangeably, causing confusion. Furthermore, the workflow for each model was not always explicitly described, requiring us to make educated assumptions. Secondly, the structure of the provided code presented difficulties. The repository contained large dependencies and implementations from other papers, which made it challenging to navigate and understand the relevant components specific to FairCLIP.

**Communication with original authors**: We contacted the authors for clarification on specific details of their study but have not received a response at the time of writing. Consequently, we proceeded with our best interpretation of their methodology and presented our findings based on the available information.

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

## Appendix

## A    Sinkhorn Distance

The Sinkhorn distance measures the discrepancy between two distributions $\mathcal{D}_B$ and $\mathcal{D}_{B_a}$ and is defined as:

$$\mathcal{W}_\epsilon(\mathcal{D}_B, \mathcal{D}_{B_a}) = \inf_{\gamma \in \Gamma(\mathcal{D}_B, \mathcal{D}_{B_a})} \left\{ \mathbb{E}_{(x,y) \sim \gamma}[c(p,q)] + \epsilon H(\gamma \mid \mu \otimes \nu) \right\}. \tag{7}$$

Here, $\Gamma(\mathcal{D}_B, \mathcal{D}_{B_a})$ represents the set of joint distributions whose marginal distributions correspond to $\mathcal{D}_B$ and $\mathcal{D}_{B_a}$. The function $c(p,q)$ is the transport cost between points $p$ and $q$ belonging to $\mathcal{D}_B$ and $\mathcal{D}_{B_a}$, respectively. The term $H(\gamma \mid \mu \otimes \nu)$ represents the relative entropy of $\gamma$ in relation to the product measure $\mu \otimes \nu$. For further details on this formulation, see Luo et al. (2024a).

## B    Word Counts

Table 7 shows the count of different words identifying demographic groups in all summarized clinical notes. Since a text can contain other words indicating demographic information (like the words "man"/"woman" for gender male/female), we randomly sampled 50 summarized notes and collected such indirect identifying words. We found that for attributes other than gender, different identifying words were not present in the text. Additionally, references to the language "Others" (one of the three language options) are uncountable, so this is not in the table.

| Identifier | Count |
|---|---|
| Asian, Black, White | 3, 13, 56 |
| Male, Female | 2166, 2450 |
| Non-hispanic, Hispanic | 653, 32 |
| English, Spanish | 0, 2 |
| **Indirect Identifier** | |
| his, her, woman, man, he, she | 332, 475, 151, 44, 465, 891 |

Table 7: Counts of attribute identifying words in the summarized notes.

## C    Influence of Language

In addition to the experiment presented in 3.3.5, we perform another experiment, which closely follows the approach of Luo et al. (2024a), but we use CLIP instead of BLIP2. After fine-tuning the model on the medical domain during pre-training, we evaluate its performance using two different linear probing setups: 1) linear probing on visual features and 2) linear probing on visual and textual features. The results of this experiment can be found in Table 8.

The difference between Table 8 and the results presented in Table 5 is the following. In Table 8, both the V and VL configurations are fine-tuned using the clinical note for text features. In Table 5, only the VL configuration is fine-tuned on the clinical note. For Table 8, the difference between V and VL is that for the V configuration, only the image features are mapped to the output layer in linear probing. For VL, the image and text features are stacked and together mapped to the output layer. We found the setup whose results are reported in Table 5 more valuable.

| Attribute | Features | DPD ↓ | DEOdds ↓ | AUC ↑ | ES-AUC ↑ | PD ↓ | MAD ↓ | G-AUC ↑ | | |
|---|---|---|---|---|---|---|---|---|---|---|
| | | | | | | | | Asian | Black | White |
| Race | V | **3.75** ± 1.18 | **8.85** ± 0.91 | 78.15 ± 2.99 | 73.77 ± 2.36 | 9.47 0.14 | 1.98 ± 0.59 | 80.81 ± 3.27 | 75.53 ± 2.07 | 78.8 ± 2.95 |
| | VL | 3.83 ± 0.95 | 9.87 ± 2.10 | **81.87** ± 1.79 | **77.35** ± 2.12 | **8.73** ± 2.41 | **1.95** ± 0.20 | **81.36** ± 2.98 | **78.57** ± 1.57 | **82.95** ± 1.63 |
| | | | | | | | | Female | Male | |
| Gender | V | 1.67 ± 1.44 | 6.7 ± 0.4 | 78.15 ± 2.99 | 73.72 ± 2.72 | 4.37 ± 3.06 | 3.00 ± 0.25 | 75.41 ± 2.9 | 81.41 ± 3.06 | |
| | VL | **0.33** ± 0.25 | **3.78** ± 2.54 | **81.87** ± 1.79 | **79.26** ± 2.43 | **3.78** ± 2.54 | **1.66** ± 0.85 | **80.35** ± 2.13 | **83.66** ± 1.80 | |
| | | | | | | | | Non-Hispanic | Hispanic | |
| Ethnicity | V | 13.53 ± 3.55 | 19.3 ± 1.74 | 78.15 ± 2.99 | **74.03** ± 4.09 | 19.26 ± 2.18 | **2.84** ± 1.15 | 78.43 ± 2.92 | **72.75** ± 4.78 | |
| | VL | **10.74** ± 6.70 | **16.59** ± 10.21 | **81.87** ± 1.79 | 73.75 ± 2.94 | **16.59** ± 10.21 | 5.53 ± 1.43 | **82.28** ± 1.72 | 71.22 ± 3.84 | |
| | | | | | | | | English | Spanish | Others |
| Language | V | 15.62 ± 3.85 | **21.67** ± 12.5 | 78.15 ± 2.99 | **69.54** ± 2.27 | 14.44 ± 3.61 | **4.15** ± 1.97 | 78.56 ± 3.08 | **80.3** ± 6.33 | **68.88** ± 2.18 |
| | VL | **13.62** ± 2.72 | 23.43 ± 5.63 | **81.87** ± 1.79 | 65.39 ± 7.99 | **9.99** ± 5.47 | 8.78 ± 4.72 | **82.59** ± 1.68 | 70.83 ± 11.51 | 67.29 ± 5.51 |

Table 8: Results of linear probing on image features only (V) and image and text features (VL) of medical fine-tuned CLIP (ViT-L/14), reporting the mean and standard deviation across three random seeds.

# D   Testing Significance of Group Difference

| Attribute | Model | P-values |
|---|---|---|
| Race | CLIP | $2.23 \times 10^{-5}$ |
| | CLIP-FT | $1.20 \times 10^{-2}$ |
| Gender | CLIP | $1.95 \times 10^{-5}$ |
| | CLIP-FT | $2.35 \times 10^{-3}$ |
| Ethnicity | CLIP | $4.51 \times 10^{-4}$ |
| | CLIP-FT | $5.07 \times 10^{-2}$ |
| Language | CLIP | $3.47 \times 10^{-7}$ |
| | CLIP-FT | $2.45 \times 10^{-2}$ |

Table 9: P-values of ANOVA test for CLIP and CLIP-FT, testing the significance of the group-wise AUC values for each attribute being different in Table 1 (linear probing).

| Attribute | Model | P-values |
|---|---|---|
| Race | CLIP B/16 | $8.71 \times 10^{-3}$ |
| | CLIP L/14 | $6.89 \times 10^{-3}$ |
| Gender | CLIP B/16 | $8.42 \times 10^{-3}$ |
| | CLIP L/14 | $4.58 \times 10^{-3}$ |
| Ethnicity | CLIP B/16 | $7.47 \times 10^{-2}$ |
| | CLIP L/14 | $7.37 \times 10^{-2}$ |
| Language | CLIP B/16 | $8.17 \times 10^{-2}$ |
| | CLIP L/14 | $2.37 \times 10^{-1}$ |

Table 10: P-values of ANOVA test for both architectures of CLIP. Testing the significance of the group-wise AUC values for each attribute being different in Table 2 (zero-shot).

# E   Testing Significance of Model Comparisons

When evaluating whether a certain model outperforms another model, we conducted one-sided independent t-tests. The test is one-sided because we aim to determine if a model achieves significantly better results across different metrics. Therefore, the null hypothesis ($H_0$) states that model X does not outperform model Y or performs the same and the alternative hypothesis ($H_1$) asserts that model X performs significantly better.

$$H_0 : \mu_{\text{X}}^{metric} \leq \mu_{\text{Y}}^{metric} \qquad H_1 : \mu_{\text{X}}^{metric} > \mu_{\text{Y}}^{metric}$$

We performed t-tests to compare CLIP and CLIP-FT, in which we tested whether the metrics of CLIP-FT were significantly higher than those of CLIP. The results of this analysis are presented in Table 11.

Additionally, we performed t-tests to compare the metrics of FairCLIP against CLIP. The alternative hypothesis states that the metrics of FairCLIP outperform those of CLIP. The results of this comparison are shown in Table 12.

| Attribute | DPD | DEOdds | AUC | ES-AUC | PD | MAD | Group-wise AUC | | |
|---|---|---|---|---|---|---|---|---|---|
| | | | | | | | Asian | Black | White |
| Race | $9.53 \times 10^{-1}$ | $9.78 \times 10^{-1}$ | $3.50 \times 10^{-1}$ | $2.54 \times 10^{-1}$ | $9.78 \times 10^{-1}$ | $7.02 \times 10^{-1}$ | $3.68 \times 10^{-1}$ | $1.79 \times 10^{-1}$ | $3.25 \times 10^{-1}$ |
| | | | | | | | Female | Male | |
| Gender | $1.75 \times 10^{-1}$ | $3.19 \times 10^{-2}$ | $3.50 \times 10^{-1}$ | $2.86 \times 10^{-1}$ | $7.35 \times 10^{-1}$ | $9.14 \times 10^{-1}$ | $3.07 \times 10^{-1}$ | $3.99 \times 10^{-1}$ | |
| | | | | | | | Non-Hispanic | Hispanic | |
| Ethnicity | $8.99 \times 10^{-1}$ | $1.63 \times 10^{-1}$ | $3.50 \times 10^{-1}$ | $2.56 \times 10^{-1}$ | $1.63 \times 10^{-1}$ | $8.61 \times 10^{-1}$ | $3.38 \times 10^{-1}$ | $2.38 \times 10^{-1}$ | |
| | | | | | | | English | Spanish | Others |
| Language | $4.36 \times 10^{-1}$ | $8.47 \times 10^{-1}$ | $3.50 \times 10^{-1}$ | $7.45 \times 10^{-1}$ | $6.91 \times 10^{-2}$ | $2.11 \times 10^{-1}$ | $2.93 \times 10^{-1}$ | $7.48 \times 10^{-1}$ | $9.71 \times 10^{-1}$ |

Table 11: One-sided t-test: p-values for CLIP versus CLIP-FT for all fairness metrics.

| Attribute | Arch. | DPD | DEOdds | AUC | ES-AUC | PD | MAD | Group-wise AUC | | |
|---|---|---|---|---|---|---|---|---|---|---|
| | | | | | | | | Asian | Black | White |
| Race | B-16 | $9.88 \times 10^{-1}$ | $9.96 \times 10^{-1}$ | $3.92 \times 10^{-1}$ | $1.37 \times 10^{-1}$ | $9.96 \times 10^{-1}$ | $8.42 \times 10^{-1}$ | $5.69 \times 10^{-1}$ | $5.80 \times 10^{-1}$ | $3.09 \times 10^{-1}$ |
| | L-14 | $3.44 \times 10^{-2}$ | $4.24 \times 10^{-2}$ | $8.49 \times 10^{-1}$ | $9.09 \times 10^{-1}$ | $7.43 \times 10^{-2}$ | $2.83 \times 10^{-1}$ | $8.62 \times 10^{-1}$ | $7.57 \times 10^{-1}$ | $8.67 \times 10^{-1}$ |
| | | | | | | | | Female | Male | |
| Gender | B-16 | $8.29 \times 10^{-1}$ | $8.80 \times 10^{-1}$ | $6.35 \times 10^{-1}$ | $6.94 \times 10^{-1}$ | $8.80 \times 10^{-1}$ | $3.12 \times 10^{-1}$ | $6.84 \times 10^{-1}$ | $5.92 \times 10^{-1}$ | |
| | L-14 | $3.84 \times 10^{-2}$ | $2.28 \times 10^{-1}$ | $9.46 \times 10^{-1}$ | $8.79 \times 10^{-1}$ | $2.37 \times 10^{-1}$ | $8.27 \times 10^{-1}$ | $9.04 \times 10^{-1}$ | $9.79 \times 10^{-1}$ | |
| | | | | | | | | Non-Hispanic | Hispanic | |
| Ethnicity | B-16 | $9.12 \times 10^{-1}$ | $7.08 \times 10^{-1}$ | $4.30 \times 10^{-1}$ | $7.19 \times 10^{-1}$ | $7.08 \times 10^{-1}$ | $3.00 \times 10^{-1}$ | $4.21 \times 10^{-1}$ | $9.85 \times 10^{-1}$ | |
| | L-14 | $7.91 \times 10^{-1}$ | $8.16 \times 10^{-1}$ | $4.87 \times 10^{-1}$ | $2.32 \times 10^{-1}$ | $8.17 \times 10^{-1}$ | $8.89 \times 10^{-1}$ | $5.16 \times 10^{-1}$ | $1.93 \times 10^{-1}$ | |
| | | | | | | | | English | Spanish | Others |
| Language | B-16 | $4.09 \times 10^{-1}$ | $6.69 \times 10^{-1}$ | $4.31 \times 10^{-1}$ | $9.70 \times 10^{-1}$ | $2.03 \times 10^{-1}$ | $4.42 \times 10^{-2}$ | $4.04 \times 10^{-1}$ | $9.58 \times 10^{-1}$ | $8.94 \times 10^{-1}$ |
| | L-14 | $9.56 \times 10^{-1}$ | $4.50 \times 10^{-1}$ | $8.30 \times 10^{-1}$ | $8.93 \times 10^{-1}$ | $4.84 \times 10^{-1}$ | $3.58 \times 10^{-1}$ | $8.17 \times 10^{-1}$ | $8.52 \times 10^{-1}$ | $8.84 \times 10^{-1}$ |

Table 12: One-sided t-test: p-values for CLIP versus FairCLIP per architecture per attribute.

