# OpenReview forum: "Reproducibility Study of "FairCLIP: Harnessing Fairness in Vision-Language Learning""
_TMLR — Rejected by TMLR_

### Review · Reviewer_H6D9 · 2025-03-13

**Summary Of Contributions:**

This paper studies the fairness of vision-language models, e.g., CLIP and BLIP2. The main contribution of this paper is to reproduce the key findings of FairCLIP, and conduct more evaluations of fairness analysis. The authors claim that the improvement of FairCLIP over CLIP are inconsistent,  and make a detailed analysis of the source of bias.
This paper potentially reveal  some novel insights of bias on medical applications.

**Audience:**

Yes

**Broader Impact Concerns:**

- Data privacy and data leakage problem of the special medical domain.
- Vulnerability of model prediction and reliable inference.

**Claims And Evidence:**

Yes

**Requested Changes:**

- Supplementary Experiments: more analysis of other benchmarks and the comparison of state-of-the-art methods specifically designed for Fairness improvements.
- Theoretical Refinement. A detailed fairness analysis of vision-language models is required.
- The authors claim that  bias does not originate from the summarized clinical notes, medical pre-training, group imbalance, or attribute identification. The potential causes are required to discuss.
- The results are all conducted by aualitative analysis. I think some quantitative analysis would further improve the quality.

**Strengths And Weaknesses:**

Strengths
- Fine experimental configurations and evaluations about fairness analysis of vision-language models could potential benefit the community.
- Novel findings beyond FairCLIP  would attract much interests to boost the fairness of existing vision-language models.

Weakness:
- Novelty. The reproduced process of FairCLIP seems to be limited technique contribution. It is worth noting that FairCLIP has opensourced the code at https://github.com/Harvard-Ophthalmology-AI-Lab/FairCLIP.
- ‌Limited Scope of Experiments. The paper only reproduces the results of FairCLIP, but do not include state-of-the-art methods
specifically designed for Fairness improvements.
- Experiments are limited to  Harvard-FairVLMed dataset. More benchmarks potentially improve current manuscript.
- Theoretical concerns. The theoretical insight for fairness analysis of vision-language models can further boost the quality‌ of the manuscript.

---

> ### Author Response · Authors · 2025-03-25
> **Initial questions regarding the comments, what is meant by 1) theoretical refinement and 2) quantitative addition.**
>
> Dear reviewer,
>
> We thank you for the comments. We are busy incorporating your and the other reviewers’ feedback. For now, we had some clarification questions regarding some of your comments.
>
> First, what do you mean by theoretical refinement of fairness analysis of vision-language models? Do you mean we should include a theoretical background of earlier work about fairness? Or a background of CLIP and bias in CLIP?
>
> Second, your final point states that adding quantitative analysis would further improve the quality. What is meant by this, given that the bulk of our analysis is based on numerical evaluation of metrics and statistical tests?
>
> Best,
> The authors

---

> ### Author Response · Authors · 2025-03-27
> **Responding to reviewer**
>
> Dear reviewer,
>
> We thank you for the feedback. Below, we address the points raised in the review.
>
> **Limited scope & novelty**
>
> Reviewer comment: This work could benefit from more analysis of other benchmarks and different state-of-the-art methods to improve fairness. Furthermore, it is of limited technical novelty, as the code of the original paper is publicly available.
>
> Response: According to the FairCLIP paper, Harvard-FairVLMed, their released dataset, is the first medical vision-language dataset that contains detailed demographic information to facilitate a fairness analysis. To the best of our knowledge, the dataset is indeed the first and only dataset suitable in the medical vision-language fairness context. We were thus unable to run our experiments on other benchmarks. Furthermore, we believe an extension to other benchmarks is appropriate to see if a model generalizes. However, since we found that FairCLIP does not consistently improve fairness on this dataset, we see little direct motivation to test it on other benchmarks.
>
> As for the next two points, to include other state-of-the-art methods and our limited technical novelty, the main goal of our work was reproducibility. We found it valuable to first examine the proposed method in detail, before implementing other methods or pursuing further technical novelty.
>
> **Theoretical insight**
>
> Reviewer comment: This work could benefit from theoretical insights regarding fairness analysis of vision-language models.
>
> Response: We interpret this as adding more theoretical background of bias in CLIP. We point out that our experiments to find the source of bias are based on earlier work. In acknowledging your comment, we have added a more thorough discussion of alternative bias mitigation strategies in Section 5.2.
>
> **Discuss source of bias**
>
> Reviewer comment: Discuss the potential source of bias.
>
> Response: We acknowledge this comment and have added a discussion of investigating the source of bias for future work in Section 5.2.
>
> **Quantitative analysis**
>
> Reviewer comment: Since all results are based on qualitative analysis, the work would benefit from adding quantitative analysis.
>
> Response: We kindly point out that most of our results are based on numerical evaluation of metrics and statistical tests. We believe this is a quantitative analysis. A qualitative analysis would typically include case studies or interpretability methods to explore, for instance, why the model performs well in certain contexts. However, we believe that this was not the intention of the reviewer’s comment, and such an analysis was not a priority for our current research.
>
> We hope that these revisions address the reviewer's concerns and improves the quality of our work. Thank you for the constructive feedback and the opporunity to strengthen our work.
>
> Best,
> The authors

---

### Review · Reviewer_wEi5 · 2025-03-13

**Summary Of Contributions:**

This study reproduces and critically examines FairCLIP: Harnessing Fairness in Vision-Language Learning. Key contributions include:
1. Reproducing Results: Confirms most findings but challenges FairCLIP’s fairness improvements.
2. Extended Fairness Evaluation: Introduces Precision Disparity (PD) and Mean Absolute Deviation (MAD) for a more nuanced assessment.
3. Bias Source Analysis: Finds no single cause of bias in CLIP, testing clinical notes, medical pre-training, and group imbalance.
4. Reproducibility Insights: Highlights ambiguities in methodology and dataset.

**Audience:**

Yes

**Broader Impact Concerns:**

1. Misuse of metrics like ES-AUC may mislead future research.
2. Addressing bias is crucial to prevent healthcare disparities.

**Claims And Evidence:**

Yes

**Requested Changes:**

1. The paper critiques FairCLIP’s inconsistent fairness improvements but does not suggest alternative bias mitigation strategies.
3. The authors exclude BLIP2 due to computational constraints, but its absence leaves a gap in fully verifying the original paper’s claims.
4. Please provide more details  on data demographic distributions

**Strengths And Weaknesses:**

Strengths:
1. Rigorous reproducibility verification.
2. Improved fairness assessment with new metrics.
3. Systematic bias investigation.


Weaknesses:
1. Excludes BLIP2 due to computational limits.
2. No new method proposed.
3. Focuses only on glaucoma detection.

---

> ### Author Response · Authors · 2025-03-27
> **Responding to reviewer**
>
> Dear reviewer,
>
> We thank you for the feedback. Below, we address the points raised in the review.
>
> **No new method proposed**
>
> Reviewer comment: No alternative bias mitigation strategies are proposed.
>
> Response: The primary focus of our work was reproducibility. We prioritized a detailed examination of the proposed method to assess its validity before proposing new approaches. Given that we were unable to confirm the main claim of the original paper, we concentrated on thoroughly analyzing the factors contributing to this outcome. By ensuring a trustworthy and transparent fairness analysis, we aim to support the development of more robust and equitable models in future studies. Thus, acknowledging the spirit of your comment, we have included a more thorough discussion of alternative bias mitigation strategies for future research in Section 5.2.
>
> **Exclusion of BLIP2**
>
> Reviewer comment: Leaving out BLIP2 due to computational constraints leaves a gap in verifying the original claims.
>
> Response: We indeed leave out BLIP2 due to computational constraints, but we point out that BLIP2 is only a minor part of the original paper’s claims. The authors use both CLIP and BLIP2 to establish that vision-language models exhibit bias. However, the argument of an improvement in fairness of the proposed method, FairCLIP, is solely based on a comparison with CLIP. Granted, BLIP2 is used in some of the ablation studies, but the center of gravity of the paper lies in the comparison between FairCLIP and CLIP.  Additionally, we note that CLIP clearly outperforms BLIP2 in the original paper’s section of establishing bias of vision-language models. Taking this all together, we saw the reproducibility of BLIP2 not as an integral part of our analysis.
>
> **Details on demographic distribution**
>
> Reviewer comment: Provide more details on the distribution of demographic data.
>
> Response: We acknowledge this comment and have added visual aids on page 3 that better show the distribution of demographic data.
>
> **Focus only on glaucoma detection**
>
> Reviewer comment: The focus is only on glaucoma detection.
>
> Response: We point out that an extension to other tasks is limited by the lack of suitable, alternative benchmarks. According to the original paper, their released dataset, Harvard-FairVLMed, is the first medical vision-language dataset that contains detailed demographic information to facilitate a fairness analysis. To the best of our knowledge, the dataset is indeed the first and only dataset suitable in the medical vision-language fairness context. We are thus unable to run the experiments on other tasks. Furthermore, we believe an extension to other tasks is appropriate to see if a model generalizes. However, since we found the fairness improvement to be inconsistent on this task, we see little direct motivation to test it on other benchmarks.
>
> We hope that these revisions address the reviewer's concerns and improves the quality of our work. Thank you for the constructive feedback and the opporunity to strengthen our work.
>
> Best, The authors

---

### Review · Reviewer_BoP3 · 2025-03-13

**Summary Of Contributions:**

This paper conducts extensive experiments to reproduce the results in “FairCLIP: Harnessing Fairness in Vision-Language Learning”. It has performed more fairness analysis by incorporating two extra metrics, namely Precision Disparity (PD) and Mean Absolute Deviation (MAD), as the complementary measurement of DEOdds and ES-AUC, respectively. Additional experiments are also conducted to investigate the potential source of CLIP’s bias, such as the impact of clinical notes, medical pre-training and group imbalance.

**Audience:**

No

**Broader Impact Concerns:**

In my own opinion, there is no ethical concerns.

**Claims And Evidence:**

No

**Requested Changes:**

Please see the weakness in the above question.

In addition, adding more figures for illustration will make this paper more readable, and incorporating more qualitative analysis (such as Grad-CAM visualizations) will make the experimental resutls more convincing.

**Strengths And Weaknesses:**

Strengths:
1. This paper is well organized, most presentations are clear and free of grammatical errors.
2. Extensive experiments are conducted, and the related experimental settings are clearly stated.
3. The usage of MAD for analyzing the fairness seems to be reasonable.

Weakness:
1. Could the authors provide the formal definition of DPD, DEOdds, and PD measurements? It will make the computational process more clear by defining them via equations.
2. This article is more like a technical report than an research paper, the authors just repeat the experiments in FairCLIP and obtain similar conclusions/results. Compared to the original paper of FairCLIP, the authors do not offer new insights about the fairness of medical vision-language learning. After reading this paper, I have not found any informative results or some inspiration for future related studies. Could the authors more clearly summarize and emphasize their new findings?
3. I am confused about the difference between DEOdds and PD. In my own opinion, if the value of DEOdds is small, it means that the TPR and FPR of different demographic groups are more similar, which will further lead to a smaller value of PD (based on the PPV defined in Eq. (3)). So could the authors explain their motivation of introducing PD measurement, and what the complementary information could this measurement provide beyond DEOdds?
4. The conclusion of “the model is not able to predict the demographic attributes from the images” made by the authors is rather arbitrary. Even the high AUC value could be caused by the group imbalance, it cannot exclude the possibility that attribute predictions can be made. Could the authors give more discussions about this?
5. The percent sign (%) is missing for AUC values of the example in “ES-AUC as Fairness Metric”, which will confuse the readers unfamiliar with AUC as to how the 1.3 in the denominator of the formula is derived.

---

> ### Author Response · Authors · 2025-03-27
> **Responding to reviewer**
>
> Dear reviewer,
>
> We thank you for the feedback. Below, we address the points raised in the review.
>
> **Formal definition of metrics**
>
> Reviewer comment: A formal definition of the metrics DPD, DEOdds and PD is missing.
>
> Response: We acknowledge this comment and have added a formal definition of these metrics in Section 3.3.4.
>
> **Lacks new insight**
>
> Reviewer comment: The article is more a technical report than a research paper, it repeats the experiments of FairCLIP and obtains similar results and conclusions. It lacks new insights about fairness in the medical vision-language context and does not give inspiration for future work. Could the new findings be more clearly emphasized?
>
> Response: Since the article is a reproducibility study, the bulk of our analysis is indeed a repetition of earlier work. We reproduced the original results but were, in fact, unable to draw the same key conclusion as the original paper, namely that FairCLIP improves fairness. We believe this underwrites the relevance of our reproducibility study.
>
> We acknowledge your comments about new insights and inspiration for future work. We have more clearly emphasized our findings in Section 1 and added a more thorough discussion of potential directions for future work in Section 5.2.
>
> **Motivation of PD and its complementarity to DEOdds**
>
> Reviewer comment: Could the authors explain their motivation of introducing PD and its complementarity to DEOdds?
>
> Response: We acknowledge this comment and have more clearly articulated our motivation for including PD and its complementarity to DEOdds to Section 3.3.4. Below, we respond directly to your comment.
>
> It is indeed true that in the case where disease prevalence is equal across demographic groups, a small DEOdds (i.e., matched TPR and FPR) would typically correspond to a small PD, as PPV is directly influenced by these values. However, this relationship breaks down when prevalence differs. When one group has higher prevalence, the same TPR and FPR values will result in higher PPV for that group, because a greater proportion of positive predictions will be true positives. Conversely, the group with lower prevalence will have a lower PPV, even if treated equally by the model in terms of detection and false alarm rates.
>
> PD therefore captures fairness in the trustworthiness of predictions, while DEOdds reflects fairness in the model’s decision behavior. This distinction is especially important in healthcare, where unequal PPV can result in unequal treatment, trust, and outcomes.
>
> **Further discuss the prediction of demographic attributes**
>
> Reviewer comment: The conclusion that the model is unable to predict the demographic attributes from the image is rather arbitrary. Could the authors give more discussion about this?
>
> Response: We acknowledge this comment and have added a better discussion of attribute prediction in Section 3.3.5 and 4.4. Now we only test and discuss the results for the gender attribute. Additionally, we refer to it under Future Research, Section 5.2.
>
> **Percent sign missing**
>
> Reviewer comment: The percent sign is missing in the section “ES-AUC as Fairness Metric”.
>
> Response: We acknowledge this comment and have made the denominator more explicit and added the percent sign in Section 4.2 under subheading “ES-AUC as Fairness Metric”.
>
> **More illustrations and qualitative analysis**
>
> Reviewer comment: More illustrations will make this paper more readable, and adding qualitative analysis will make the experimental results more convincing.
>
> Response: We acknowledge this comment and have added a flowchart of the algorithmic steps of FairCLIP, as well as other visual aids. As for the qualitative analysis, we considered adding a visual analysis like Grad-CAM but eventually decided not to pursue it. Given the nature of the data, which consists of close-up laser images of the retina taken under the same conditions, we speculated that the predictive features in the images were difficult to discern, and that the output of Grad-CAM would be difficult to interpret.
>
> We hope that these revisions address the reviewer's concerns and improves the quality of our work. Thank you for the constructive feedback and the opporunity to strengthen our work.
>
> Best, The authors

---

### Decision · Action_Editor_qigW · 2025-04-26

**Recommendation:** Reject

**Comment:**

Most reviewers for this reproducibility paper have given negative recommendations. After reviewing the revised version myself, I believe the central question for this paper, based on the TMLR acceptance criteria, is:

*Does this reproducibility paper systematically study the generalizability of a published method and lays out actionable lessons for its audience?*

To assess this, let's examine the contributions presented in the paper:
> 1.	We replicate the experiments in Luo et al. (2024a) and find that most results are reproducible.
> 2.	By evaluating the authors’ reasoning, we conclude that the key metric to claim fairness improvements, Equity-Scaled AUC, is flawed. We apply two additional metrics: one from the literature and another as a modification to this key metric. We find that an improvement in fairness of FairCLIP over CLIP is inconsistent.
> 3.	We run several experiments in an effort to determine the source of bias. We find no conclusive evidence that the clinical notes, medical pre-training, or group imbalance are fundamental sources of bias.

The first contribution, the replication of experiments, is a basic educational component of reproducibility work, as defined in the TMLR acceptance criteria.

Regarding the second contribution, the authors argued that the ES-AUC metric has an inherent limitation because it cannot reliably represent a proper trade-off curve. I believe this is a valuable point concerning the fairness metric. However, I don't think the authors adequately emphasized this contribution. Let's look closely at Section 4.2:

> To illustrate this, consider a scenario where a model’s overall performance (i.e., AUC) improves, while the disparity among subgroups remains unchanged. For instance, if all subgroup AUC values increase uniformly, ES-AUC will rise solely due to the overall performance gain, even though fairness has not improved.

This statement isn't necessarily accurate. If all subgroup AUC values improve uniformly, it means the worst-performing group is now performing better than before. From the perspective of the lowest performance, fairness has indeed improved. For example, if the performance of two subgroups goes from 0.5 to 0.8 and from 0.6 to 0.9, the fairness is significantly better because the lowest performance is now 0.8 instead of 0.5. Therefore, it's not unreasonable to make such a claim.

> More importantly, if the disparity between subgroups increases, but the overall AUC improves, the ES-AUC may still increase despite worsening fairness. For example, suppose a model initially has an overall AUC of 60% and has no disparity among its subgroups. Therefore, the ES-AUC would simply be equal to the overall AUC. If the overall AUC increases to 80%, with group-wise AUC values of 90%, 85%, and 65%, this will give: ES-AUC = 0.615

In my view, ES-AUC is a metric designed to balance performance and fairness. In the given counterexample, as I mentioned, the worst-performing group's AUC has improved by 5%. Therefore, even though the gap is 35%, the performance of the least advantaged group has indeed increased.

Furthermore, this rationale for fairness via improving the worst-performing subgroup can be supported by the philosophy of the **difference principle**. Specifically:

*Social and economic inequalities are to be arranged so that they are both (a) to the **greatest expected benefit of the least advantaged** and….*
[Citation. Rawls, John (1999). A Theory of Justice: Revised Edition. p. 72]

Therefore, I believe there are reasonable arguments to support ES-AUC. While I agree that all fairness metrics can have limitations, the key point is that current paper doesn't adequately discuss this point.

Regarding the third contribution, the investigation into the source of bias is an important area. However, I didn't fully grasp the precise experimental design used to identify these sources. This analysis seems to lack generalizable insights.

Based on all the evidence, I don't think this paper provides sufficient generalizable insights. Therefore, my decision is rejection with major revision.

**Audience:**

Partially, see the comments for details

**Claims And Evidence:**

Partially, see the comments for details

**Resubmission Of Major Revision:**

The authors may consider submitting a major revision at a later time.